# Study on In-Situ Synthesis Process of Ti–Al Intermetallic Compound-Reinforced Al Matrix Composites

**DOI:** 10.3390/ma12121967

**Published:** 2019-06-19

**Authors:** Qiong Wan, Fuguo Li, Wenjing Wang, Junhua Hou, Wanyue Cui, Yongsheng Li

**Affiliations:** 1State Key Laboratory of Solidification Processing, School of Materials Science and Engineering, Northwestern Polytechnical University, Xi’an 710072, China; wq_haust@sina.com (Q.W.); 18334703545@163.com (W.W.); junhuahou@mail.nwpu.edu.cn (J.H.); cwy1006@mail.nwpu.edu.cn (W.C.); yongshengli@mail.nwpu.edu.cn (Y.L.); 2College of Material Science and Engineering, Henan University of Science and Technology, Luoyang 471023, China; 3Shaanxi Key Laboratory of High-Performance Precision Forming Technology and Equipment, Northwestern Polytechnical University, Xi’an 710072, China

**Keywords:** intermetallic compounds, Al matrix composites, ball-milling, core–shell-like microstructure

## Abstract

In this study, ball-milled powder of Ti and Al was used to fabricate Ti–Al intermetallic compound-reinforced Al matrix composites by an in-situ reaction in cold-pressing sintering and hot-pressing sintering processes. The detailed microstructure of the Ti–Al intermetallic compound-reinforced Al composite was characterized by optical microscopy (OM), X-ray diffraction (XRD), energy dispersive spectrometry (EDS), and electron backscattered diffraction (EBSD). The results indicate that a typical core–shell-like structure forms in the reinforced particles. The shell is composed of a series of Ti–Al intermetallic compounds and has good bonding strength and compatibility with the Al matrix and Ti core. With cold-pressing sintering, the shell around the Ti core is closed, and the shell thickness increases as the milling time and holding time increase. With hot-pressing sintering, some radiating cracks emerge in the shell structure and provide paths for further diffusion of Ti and Al atoms. The Kirkendall effect, which is caused by the difference between the diffusion coefficients of Ti and Al, results in the formation of cavities and a reduction in density degree. When the quantity of the intermetallic compounds increases, the hardness of the composites increases and the plasticity decreases. Therefore, factors that affect the quantity of the reinforcements, such as the milling time and holding time, should be determined carefully to improve the comprehensive properties of the composites.

## 1. Introduction

The advantages of aluminum and its alloys include its light weight, low density, and good ductility. Al matrix composites (AMCs) are one of the most important and widely used metal matrix composites (MMCs) because of their high specific strength, good thermal conductivity, good electrical conductivity, low coefficient of thermal expansion, good endurance against fatigue, and stability at high temperatures [1,2,3]. In recent years, many researchers have developed metal matrix composites reinforced by ceramic particles, such as SiC, ZrO_2_, Al_2_O_3_, TiB_2_, B_4_C, and TiC [4,5,6,7,8,9,10]. These composites are widely used in the automobile industry and for structural applications. Compared with traditional aluminum alloys, ceramic-reinforced AMCs markedly improve the mechanical properties, but the high brittleness, low ductility, and large differences in the coefficients of thermal expansion between the matrix and ceramics have restricted the wide use of ceramic-reinforced AMCs [11,12,13]. Intermetallic compounds have many extraordinary properties owing to the special type of bond and atomic distribution state. Intermetallic compounds have a low density, high melting point, high strength, and good resistance to corrosion and creep [14,15]. Moreover, intermetallic compounds are compatible with the matrix [16]. Therefore, improving the manufacturing processes and properties of intermetallic compound-reinforced AMCs has become a research hotspot in recent years. 

The reinforced phase can be obtained by two methods during the manufacturing of MMCs: Ex-situ synthesis and in-situ synthesis. An ex-situ reinforcement is prepared separately before the synthesis of the MMC, and surface contamination decreases the interface reaction and wettability between the reinforcement and matrix. Moreover, the size and morphology of ex-situ reinforcements are restricted by the initial powder conditions [17,18]. On the other hand, an in-situ-formed reinforcement has good compatibility with the matrix. The interfaces between the reinforcement and matrix are stable and clear. As the reinforcement, small in-situ intermetallic compound particles are able to not only improve basic properties such as strength, hardness, and the elastic modulus, but also meet the plastic deformation requirement to the greatest extent possible [19,20,21,22,23,24]. 

Many researchers have studied the in situ process of synthesizing intermetallic compound-reinforced AMCs. Y Xue et al. fabricated in situ Al_3_Ni intermetallic compound-reinforced Al matrix composites through the powder metallurgy method and investigated the effect of sintering temperature on the microstructure and mechanical properties of the AMC [25]. R Maiti et al. ball-milled and sintered a mixture of aluminum and 10 wt% hydrated molybdenum oxide powder to synthesize Al matrix composites that were reinforced with molybdenum aluminide nanoparticles. The constitution of molybdenum aluminides differs as temperature varies [26]. CY Liu et al. manufactured Al_3_Mg_2_ intermetallic compound-reinforced AMCs by the accumulative roll-bonding (ARB) process. The mechanical properties and density of this composite after the sixth ARB cycle were compared with annealed and ARB monolithic 1060 Al [27]. Swiderska-Sroda et al. consolidated nanocrystalline powder, which was ball-milled from an Al-21at%Ti composite processed by conventional ingot metallurgy, by either sintering or explosive shockwave. They determined the optimum parameters of the ball-mill/sintering process on the basis of the combined performance of the elastic modulus, microhardness, and fracture toughness [28]. 

For AMCs, TiAl_3_ is the preferred reinforcement because of its high-temperature strength and low density, but the problem with TiAl_3_ is its low ductility at room temperature [29]. In TiAl_3_-reinforced AMCs, TiAl_3_ can improve the hardness and high-temperature strength of the matrix, while the good plasticity of the Al matrix can accommodate the deformation of the hard and brittle reinforcement. If the interface characteristics between the matrix and TiAl_3_ are distributed in the transition state, the AMC will exhibit excellent workability and high strength. These properties are desirable in automotive industries and engineering applications. The production of TiAl_3_-reinforced AMCs can use various casting techniques, such as compo casting, squeeze casting, stir casting, etc. [30]. The main problem in ingot metallurgy is the uncontrollability of the reaction and the inhomogeneous structure formed. However, in-situ TiAl_3_-reinforced AMCs can be produced by mechanical alloying and sintering. The performance of AMCs can be optimized by adjusting ball-milling parameters (rotation speed, milling time, the ratio of milling balls to material, and so on) and sintering parameters (temperature, holding time, and pressure), but studies on factors that affect the material properties have not been sufficiently systematic as a result of the high number of factors that need consideration. In order to study this process effectively with a small number of experiments, the milling time, sintering process (cold-pressing or hot-pressing sintering), and holding time were selected as the independent variables in this study. During the ball-milling process, the blended powder particles of pure aluminum and titanium were locked mechanically or cold-welded when the metal particles were plastically deformed by the collision of the milling balls. The as-milled powder was compacted and sintered to manufacture in-situ Ti–Al intermetallic compound-reinforced Al matrix composites. In order to determine the kinds and structures of in-situ-formed intermetallic compounds and study the effect of reinforcements on the properties of Al matrix composites, the constitution and morphology of the reinforcement were observed by optical microscopy (OM) and electron backscattered diffraction (EBSD). Thus, the effects of the selected process parameters on the microstructure of the reinforcement and the properties of the composites were analyzed in this research.

## 2. Experiment

Pure metal Ti powder (99.4% chemical purity and ~74 μm in size) and Al powder (99.0% chemical purity and ~11 μm in size) were employed to prepare the Ti–Al intermetallic compound-reinforced Al matrix composites. The blended powder of Ti and Al was mechanically milled by the lightweight horizontal planetary ball mill WXQM-4A. The molar ratio of Ti to Al was 1:10, the mass ratio of milling balls to the material was 10:1, the rotation speed was 150 rpm, and the milling times were 3, 8, and 20 h. 

Cold-pressing sintering and hot-pressing sintering were used to manufacture particle-reinforced AMCs. In the cold-pressing sintering process, a specified amount of as-milled powder was put into the die, and a cylindrical billet with a diameter of 16 mm was consolidated under a load of 20 tons by a hydraulic press. Then, the billet was put into a vacuum sintering furnace, and the temperature was raised to 460 °C at a heating rate of 10 °C/min when the degree of vacuum was below 2 × 10^−3^ Pa. The billet was held at 460 °C for 1.0 h or 1.5 h, and then the furnace was cooled to room temperature. In the hot-pressing sintering process, the graphite mold was filled with as-milled powder and put in a vacuum hot-pressing sintering furnace (YZY-40-10T, Shanghai Yuzhi Mechanical and Electrical Equipment Co., Ltd., Shanghai, China). The billet was heated to 460 °C at a rate of 10 °C/min when the degree of vacuum was below 2 × 10^−3^ Pa, and then 5 MPa of pressure was applied to the indenter. The billet was held at 460 °C for 1 h and cooled to room temperature in the furnace. X-ray diffraction (XRD) was performed first using an X-ray diffractometer (XRD, Panalytical X’Pert PRO, Malvern Panalytical, Almelo, the Netherlands) to confirm the formation of intermetallic compounds. The samples were corroded by Keller’s reagent and observed by OM (OLYMPUS PMG3, Tokyo, Japan). To further understand the constitution and structure of the in-situ-reacted intermetallic compounds, electropolished samples were investigated by energy dispersive spectrometry (EDS, Inca X-sight, Oxford Instruments, Oxford, England) and EBSD (FEI Quanta 600F, Hillsboro, OR, USA). Then, Vickers hardness was measured using an automatic hardness tester (HV-50Z, Shanghai Yanrun Light-Mach Tech Co. Ltd., Shanghai, China), and the relative density was calculated for the Ti–Al intermetallic compound-reinforced AMC. Finally, the true stress–strain curves of the composites were obtained by a compressive test on an Instron 3382 electronic universal material testing machine.

To conveniently describe the specimens prepared with different milling times and sintering process parameters, the samples are designated A, B, C, D, and E, as listed in Table 1.

## 3. Experimental Results and Analysis 

### 3.1. Milling Time

The materials prepared by mechanical alloying (MA) have a homogeneous and fine microstructure with well-distributed reinforcement, and their mechanical properties are superior to those of the same materials prepared by traditional processes. Therefore, MA is regarded as one of the most effective methods for manufacturing Ti–Al intermetallic compound-reinforced Al matrix composites. MA is a method of solid powder metallurgy in a no-equilibrium state. When the powders are ball-milled, the powders are severely deformed, welded, and broken repeatedly. Then, particles of uniform size are formed and used in cold-pressing sintering or hot-pressing sintering [31,32]. The milling time and rotation speed are two important process parameters that affect the morphology, particle size, and bonding degree of the as-milled powder. In this study, the rotation speed was constant and the milling time was selected as a process variable.

Figure 1 shows the SEM images of pre-prepared powder. Figure 1a is the aluminum powder in the initial state and has an average size of about 11 μm. Figure 1b–d are the images of the powder after it was ball-milled for 3, 8, and 20 hours, respectively. It can be seen from Figure 1b that the size of particles milled for 3 hours is uniform. Most particles are about 10 μm, and only a few particles reach approximately 30 μm, which indicates that the Ti powders with an initial size of 74 μm were broken and refined under the impact of the milling balls. A portion of the powder particles are mechanically combined, but the number of combined particles is small and the bonding strength is weak because the milling time is short. When the milling time increases to 8 hours, the quantity of combined particles markedly increases and the size becomes larger, as shown in Figure 1c. After 20 hours of milling, the size of the combined particles increases to 100–200 μm; few particles with a size of about 10 μm can be seen. Comparing the images in Figure 1 reveals that the quantity and bonding strength of the combined particles increase because there is more mechanical energy with increasing milling times. A large combined particle is composed of small powder particles that were locked mechanically or cold-welded during plastic deformation. The closely bonded powder particles easily densify and interact in the subsequent cold-pressing sintering or hot-pressing sintering. When Ti powder is wrapped by many Al powder particles, Ti–Al combined particles form. The in-situ reaction occurs around the Ti cores during sintering, and Ti–Al intermetallic compounds are produced and act as the reinforced particles in the composites. The stronger bonding between Ti and Al, which means a shorter distance between them, is favorable for atom diffusion in high temperatures and facilitates the completion of the in-situ reaction. However, increasing the milling time causes an uneven distribution of particle size and results in in-situ-formed intermetallic compounds of different types that have different microstructures and uneven distributions, all of which affect the comprehensive properties of the final AMC.

### 3.2. Consolidation and Sintering Process

Figure 2 is the XRD spectra of specimen C, and it indicates that Ti–Al intermetallic compounds, including Ti_3_Al, TiAl, and TiAl_3_, formed in the Al matrix composites. Figure 3 is the metallographic images of the Ti–Al intermetallic compound-reinforced Al matrix composites prepared by the consolidation and sintering process. The microstructures of specimen A (defined in Table 1) are shown in Figure 3a,b, in which it is observed that the interface between the Ti and Al particles is clear and that there is no transition layer between them. The results indicate that the energy absorbed in the ball-milling process is not high enough to induce a chemical reaction in the blended powder. However, the mechanical energy and distortion energy stored in the powder lead to a spontaneous in-situ reaction when the activation energy is lowered by the high temperature in the sintering process. Then, the intermetallic compounds of Ti and Al form and reinforce the Al matrix. Pronounced core–shell-like structures are observed in the reinforced particles of the AMC by OM. The dark, near-spheroid particles, which are enwrapped by bright shells, are distributed in the Al matrix. The information in Figure 4 and Figure 5 shows that the major constituent of the dark and spheroid particles is Ti, the major constituent of the matrix is Al, and the shell structure between them is a transition layer constituted by a series of intermetallic compounds of Ti and Al. At a sintering temperature of 460 °C, the in-situ reaction takes place at the interface of Ti and Al. The Al atoms diffuse into the Ti particles because the diffusion coefficient of Al is higher than that of Ti. The intermetallic compounds TiAl_3_, TiAl_2_, TiAl, and Ti_3_Al form in sequence in the diffusing direction of Al, and finally, a shell structure is built that separates the Ti particle from the Al matrix. As the reaction proceeds, the thickness of the shell increases and the diffusion resistance becomes large, which slows the reaction rate. The outer layer of the shell structure is composed of TiAl_3_. TiAl_3_, which is one kind of intermetallic compound formed from Ti and Al, is rich in Al content. Compared with the traditional ceramic particles and ex-situ reinforcement, the TiAl_3_ layer has a high bonding strength and excellent compatibility with the Al matrix. Conversely, the inner-layer Ti_3_Al is particularly rich in Ti. The interface between the Ti_3_Al layer and the Ti core is clear and favorable. The transition layer of TiAl, which has relatively good plasticity among the intermetallic Ti–Al compounds, is formed between the outer and inner layers. This transition layer is conducive to the deformation during plastic processing, which relieves the cracking of the shell structure and the breakaway between layers. The diffusion degree and distance, which are affected by the milling time and sintering process, lead to changes in the shell structure and constituents and affect the mechanical properties of the final composites. 

A comparison of the images in Figure 3c–f indicates that the shell thickness of specimens produced with 8 h ball-milling is larger than the ones produced with 3 h ball-milling in the same cold-pressing sintering conditions. This is because as the milling time increases, the absorbed energy of the prepared powder becomes larger and the bonding between particles is stronger, and these properties promote diffusion during sintering and facilitate the completion of the in-situ reaction. Therefore, the shell from the 8 h ball-milling is thicker. As shown in Figure 3c,d,g,h, both specimens were milled for 3 h, but the intermetallic compound shells around the Ti cores become markedly thicker when the holding time during sintering increases. The in-situ intermetallic compounds, which are generated around adjacent Ti particles, extend and form a small connected region. The dramatic increase in intermetallic compounds enhances the strength and hardness of the composites, but it simultaneously causes inhomogeneity of the structure and properties. The accumulation of hard and brittle reinforcement results in worsened plasticity. In order to solve this problem, severe plastic deformation (SPD) and thermomechanical treatment can be performed to realize the optimum combination of strength and plasticity. In contrast to Figure 3d, bright rod-like structures are found in Figure 3f,h through careful observation. There are two reasons that explain this phenomenon. First, Al atoms with high diffusibility and diffusion rates enter Ti cores and react in-situ with the surrounding Ti; thus, rod-like intermetallic compounds form in the cores of Ti. Second, the blended powder particles of Ti and Al are welded and broken repeatedly from the impact of the milling balls, and some Ti particles may envelop some Al powder when plastic deformation occurs during milling. The Al powder in the Ti core reacts in-situ with the surrounding Ti during the sintering process and forms the observed bright intermetallic compounds. For the same milling time of 8 h, the shell thickness of the specimen prepared by hot-pressing sintering is larger than that prepared by cold-pressing sintering, as shown in Figure 3e,f,i,j. The main difference is the closed shell structure formed in cold-pressing sintering and the radiating cracks that run through the shell structure in hot-pressing sintering. At the initial compression stage of hot-pressing sintering, the distance between particles becomes shorter and the diffusibility of atoms becomes stronger from the combined action of temperature and pressure. Therefore, the in-situ reaction happens quickly, and the shell structures composed of intermetallic compounds build up. At the sintering temperature of 460 °C, the solidified Al matrix has excellent plasticity, and compressive deformation takes place easily in the Al matrix when the hydraulic cylinder rises. However, the shell structures around the Ti cores are hard and brittle. When a little compressive deformation occurs in the shell structure, the cross-section perpendicular to the compressive direction is subjected to tensile stress, from which cracks originate along the radiating directions. The broken shells provide paths for the diffusion of Ti and Al, which favors the continuation of in-situ reactions. According to the Kirkendall effect, Al atoms with a large diffusion coefficient diffuse into Ti cores through cracks in the shells and form intermetallic compounds with Ti atoms. The positions that were initially occupied by the Al atoms cannot be filled by other surrounding atoms; thus, cavities emerge, as shown in Figure 3i,j.

Figure 4 shows that the diffusion range of Ti in specimen E is larger than that in specimen C. The inferior activity of Ti in the cold-pressing sintering process slows the formation of intermetallic compounds and results in a thin shell. Conversely, the diffusibility of Ti during the hot-pressing sintering process not only thickens the shell in the reinforcement but also promotes the diffusion of some individual Ti atoms into the Al matrix to form small solid intermetallic compounds. Similarly, the quantity of Al in the Ti core is larger in specimen E than that in specimen C. This indicates that there are more intermetallic compounds formed in the core. In Figure 4b, the quantities of both Al and Ti approach zero in a small segment of the scan line, and this segment corresponds to the cavities formed during the hot-pressing sintering process. 

As shown in Figure 5, the selected area is Ti-rich, and lots of intermetallic compounds are formed in-situ in this area. These compounds can be divided into two types. One lies in the large core–shell-structured reinforcements; this includes the intermetallic compound shell and a small number of compounds formed in the Ti core. This structure is confirmed by the OM images shown in Figure 3e,f and the constituent linear scanning images shown in Figure 4. The other type of compounds is the small-sized solid particles distributed in the Al matrix. In this local area, a larger amount of TiAl_2_ forms because the local content of Ti is high. The large number of these compounds increases the hardness and brittleness of this area and affects the plasticity of the composites accordingly. This anisotropy in microstructure and properties caused by component segregation should be avoided by adjusting ball-milling parameters. 

### 3.3. Hardness and Density

The upper and lower surfaces of the specimens were polished carefully to ensure parallelism between them in the Vickers hardness test. In order to determine the hardness value properly, five points were selected along the diameter and measured in every specimen, and the mean values were calculated as the final hardness. The load in the measurement was 10 kg, and the holding time was 15 seconds. The indenter used in the test has a diamond head with a rectangular pyramid. The mean values of the two diagonals in the Vickers hardness measurement are between approximately 0.28 and 0.51 mm. Vickers hardness values of the specimens are shown in Figure 6.

Three conclusions can be drawn from Figure 6. First, the hardness of specimen E, which was prepared by hot-pressing sintering, is conspicuously larger than that of specimens B, C, and D, which were prepared by cold-pressing sintering. This is because more Ti–Al intermetallic compounds, especially the hard and brittle compound TiAl_3_, form during hot-pressing sintering. As a result, the hardness of this composite is high. Among specimens B, C, and D prepared by cold-pressing sintering, the difference in hardness between B and C is small, while the hardness of specimen D increases by 14.78% compared with that of specimen B. This shows that the holding time plays a more important role than the milling time in the formation of intermetallic compounds. Second, the hardness of specimen D, which was produced with a long holding time, is larger than that of specimen C, which was produced with a short holding time. When the holding time of the sintering process increases, the quantity of intermetallic compounds increases and their distribution widens. Consequently, the hardness of the composite improves. Third, the hardness of specimen A, produced without sintering, is the largest because the cold work hardening plays an important role when the Ti and Al particles are plastically deformed under hydraulic pressure. While the intermetallic compounds form in the subsequent sintering process reinforce the matrix, the work hardening and stress concentration are relieved at high temperatures, rendering the composites soft after sintering.

The sintering densification *K* is one aspect of a material’s porosity and equals the ratio of the actual density *ρ_a_* to the theoretical density *ρ_t_* of the specimen. The theoretical density *ρ_t_* of the composites was calculated by the rule of mixtures, and the actual density of the composites was measured on the basis of Archimedes’ principle. In order to ensure measurement accuracy, every specimen was measured five times, and the final result is the mean value of the five values, as shown in Table 2.

The sintering densification *K* was determined for each specimen and is listed in Table 3.

It can be seen from Table 3 that the relative density of the three specimens produced by cold-pressing sintering (samples B, C, and D) is almost constant and remains at about 88.5% (mean value). On the other hand, the relative density of the specimen prepared by hot-pressing sintering is relatively low because the pressure applied is small and more cavities form under the influence of the Kirkendall effect. The existence of the Kirkendall effect in the diffusion reaction of solid Ti and Al has been verified by many researchers [33,34,35]. In hot-pressing sintering, the radiating cracks destroy the integrity of the shell and promote the diffusion of Al. Thus, the Kirkendall effect in hot-pressing sintering is more pronounced than that in cold-pressing sintering.

### 3.4. Compression Performance 

In order to obtain the compression performance of the Ti–Al intermetallic compound-reinforced Al matrix composites, the specimens were wire-cut into cylinders with an 8 mm diameter and 12 mm height. Specimen E was broken during the wire cutting and unable to undergo the compression test because a large number of cavities formed during the hot-pressing sintering process. The true stress–true strain curves are shown in Figure 7.

Compared with the pure Al sample manufactured by powder metallurgy, the breaking strains of the composites decrease to different degrees, but the products of strength and ductility of these composites are higher for the same strain. This indicates that the composites have high ductility and toughness. It can be seen that the strength and plasticity of sample B are inferior to those of sample C because the long milling time of sample C contributes to the formation of intermetallic compounds and results in an improvement in strength. The strength of sample D is larger than that of sample C because the long holding time makes the in-situ reaction proceed more thoroughly. As a result, the number of reinforcement compounds increases, and their distribution range extends. As the reaction proceeds, the outer shells around some of the dispersed Ti particles enlarge and connect in a small region. The intermetallic compounds in one connected region act as significant reinforcement in the Al matrix and may cause brittle fractures in the compression test. Comparison of the compression curves of samples B, C, and D reveals that the holding time in the sintering process has a more significant effect on the ductility of the composites. Therefore, an optimum process and its corresponding process parameters, which should balance the strength and plasticity of the composites, should be determined by further study.

## 4. Conclusions

In this study, the blended powder of Ti and Al was ball-milled and fabricated in situ to form Ti–Al intermetallic compound-reinforced Al matrix composites using the processes of cold-pressing sintering or hot-pressing sintering. From the analysis of the microstructures and properties, four conclusions are obtained.
(1)A typical core–shell-like structure is produced in the Ti–Al intermetallic compound-reinforced Al matrix composites prepared by ball-milling and in-situ reaction. The outer layer of the shell structure is TiAl_3_, and the inner layer is Ti_3_Al; these species have good bonding strength and compatibility with the Al matrix and Ti core, respectively.(2)The shell is composed of a series of Ti–Al intermetallic compounds. In cold-pressing sintering, the shell thickness increases with increased milling time in ball-milling and holding time in sintering. In hot-pressing sintering, the combined action of pressure and temperature induces an in-situ reaction. The Ti–Al intermetallic compounds around the adjacent Ti cores extend and form a small connecting region.(3)The closed shell structure formed in cold-pressing sintering and the radiating cracks running through the shell structure in hot-pressing sintering are the major differences in the structure of the reinforcement. The radiating cracks that occur in hot-pressing sintering provide paths for further diffusion and enhance the Kirkendall effect, which results in more cavities and decreases the density degree.(4)From the results of this study, the holding time has a more significant effect on the hardness and ductility of the fabricated composites than the milling time.

## Figures and Tables

**Figure 1 materials-12-01967-f001:**
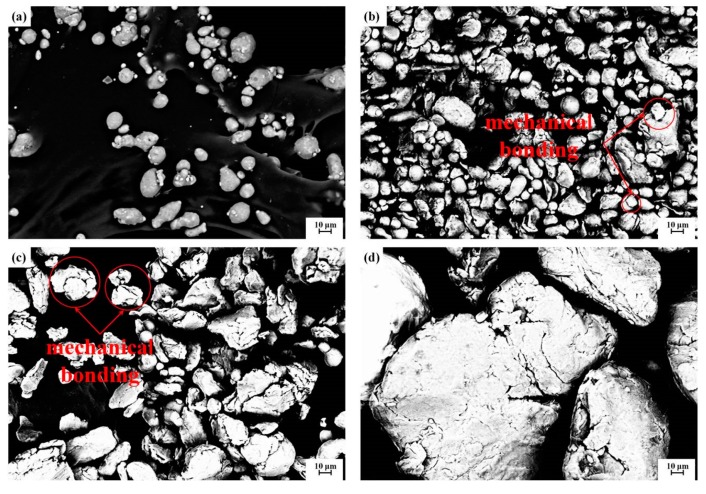
SEM images of the starting powder. (**a**) Raw Al powder, (**b**) blended powder after 3 h of ball-milling, (**c**) blended powder after 8 h of ball-milling, and (**d**) blended powder after 20 h of ball-milling.

**Figure 2 materials-12-01967-f002:**
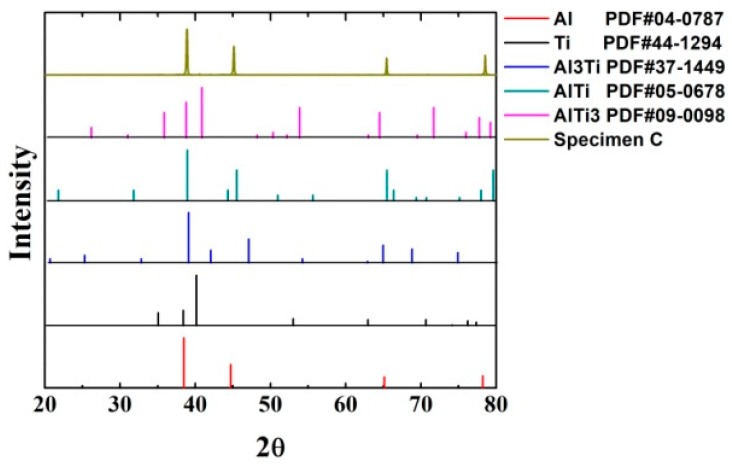
XRD spectra of specimen C.

**Figure 3 materials-12-01967-f003:**
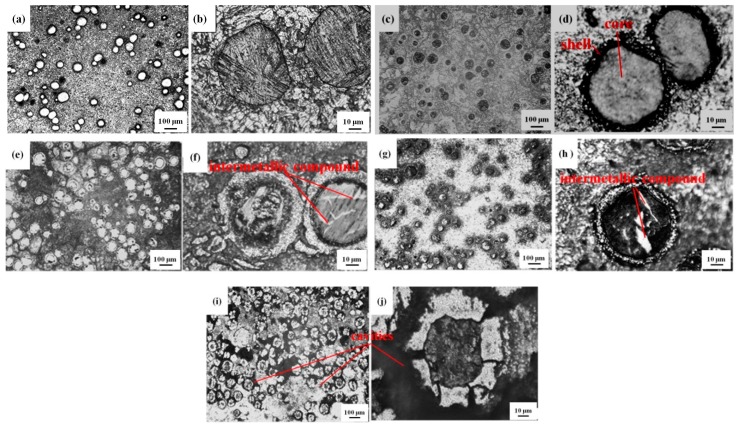
Metallographic images of AMC specimens. (**a**), (**b**) Specimen A; (**c**), (**d**) specimen B; (**e**), (**f**) specimen C; (**g**), (**h**) specimen D; (**i**), (**j**) specimen E.

**Figure 4 materials-12-01967-f004:**
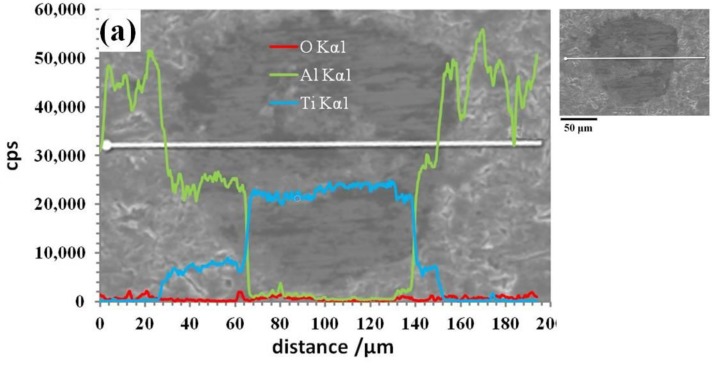
Constituent linear scanning images of specimens. (**a**) Specimen C, (**b**) specimen E.

**Figure 5 materials-12-01967-f005:**
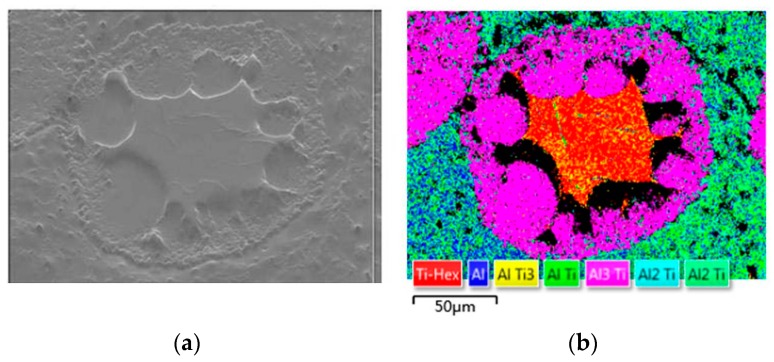
(**a**) SEM image and (**b**) EBSD analysis of specimen C.

**Figure 6 materials-12-01967-f006:**
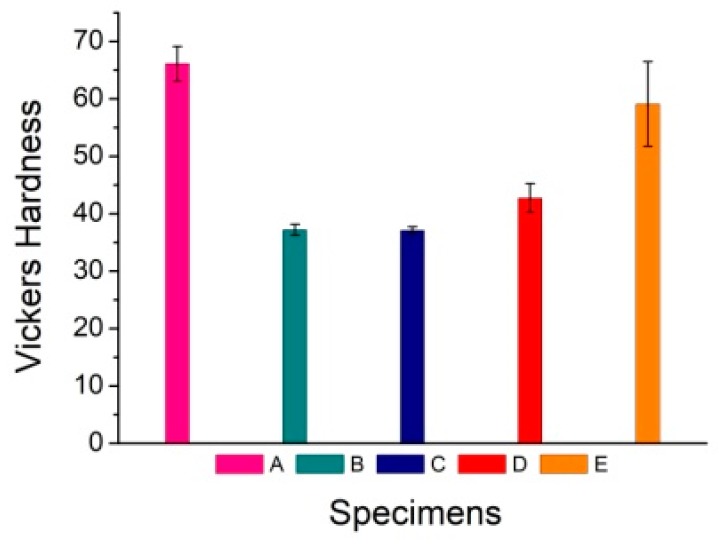
Vickers hardness of specimens.

**Figure 7 materials-12-01967-f007:**
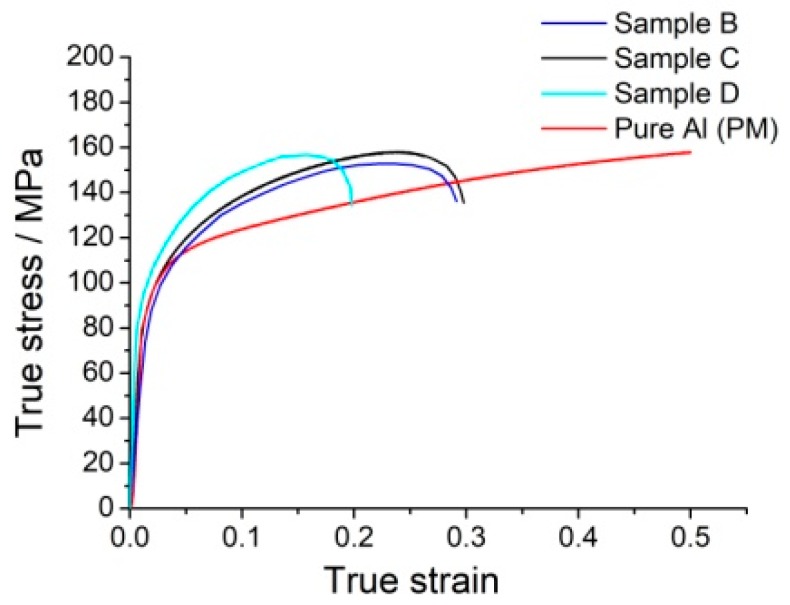
Compression curves of samples.

**Table 1 materials-12-01967-t001:** Specimens prepared with different milling times and sintering process parameters.

Sample Mark	Mill Time (h)	Consolidation and Sintering Process	Holding Time in Sintering (h)
A	3	cold pressing	/
B	3	cold-pressing sintering	1.0
C	8	cold-pressing sintering	1.0
D	3	cold-pressing sintering	1.5
E	8	hot-pressing sintering	1.0

**Table 2 materials-12-01967-t002:** Measured values and mean values of density.

Specimens	Measured Value					Mean Value
B	2.514	2.435	2.623	2.605	2.464	2.528
C	2.543	2.456	2.598	2.615	2.514	2.545
D	2.561	2.486	2.551	2.611	2.538	2.549
E	2.373	2.366	2.382	2.410	2.392	2.384

**Table 3 materials-12-01967-t003:** Sintering densification *K* of specimens.

Specimens	Sintering Densification
B	88.1%
C	88.7%
D	88.8%
E	83.1%

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
