# Peer review of "Study on In-Situ Synthesis Process of Ti–Al Intermetallic Compound-Reinforced Al Matrix Composites"

_materials, 2019, doi:10.3390/ma12121967_

Reviewer 1 Report

Dear Editor:  I would like to express my deep thanks for inviting me to review the manuscript ID: materials-517469

Title: Study on in-situ synthesis process of Ti-Al intermetallic compounds reinforced Al matrix composites with core-shell-like structure

Authors: Wan Qiong, Li Fuguo, Wang Wenjing, Hou Junhua, Cui Wanyue, Li Yongsheng

Comments:

Title: Study on in-situ synthesis process of Ti-Al intermetallic compounds reinforced Al matrix composites with core-shell-like structure

Replaced by

Study on in-situ synthesis process of Ti-Al intermetallic compounds reinforced Al matrix composites

Abstract:

-----Ti-Al intermetallic compounds reinforced aluminum matrix composites

Replaced by

---Ti-Al intermetallic compounds reinforced aluminum (Al) matrix composites---"

After that Please replaced throughout the manuscript aluminum to Al

----“ The specimens were observed to study the microstructure of reinforcement by OM, EDS and EBSD. -”

Replaced by

---- A detail microstructural characterization of Ti-Al intermetallic compounds reinforced Al composite was carried out by OM, EDS and EBSD

Introduction part:

“In recent years, many researchers developed ceramic particle reinforced aluminum matrix composites, which were used in the automobile industry and manufacture of other structural parts. The widely used ceramic particles are SiC, Al2O3, TiB2, B4C and TiC [4-8]”

Replaced by

“In recent years, many researchers developed ceramic particle reinforced metal matrix composite, which were widely used in the automobile industry and structural applications. The widely used ceramic particles are SiC, ZrO2, Al2O3, TiB2, B4C and TiC -----[add ref.]”

1. A.K. Gain, L Zhang, Microstructure, mechanical and electrical performances of zirconia nanoparticles-doped tin-silver-copper solder alloys, J. Mater. Sci.: Mater. Electron. 27(7), (2016) 7524-7533.

2.  A.K. Gain, L Zhang, M.Z. Quadir, “Composites matching the properties of human cortical bones: the design of porous titanium-zirconia (Ti-ZrO2) nanocomposites using polymethyl methacrylate powders” Mater. Sci. Eng. A 662 (2016) 258-267.

Please add references “The intermetallic compounds have low density, high melting point, high strength and good resistance to corrosion and creep.”

Please add the novelty and aim of this work in page 2 line 91-93

Experimental procedure part:

Please include all equipment name and model number.

Results and discussion:

(i) Figure 1: Please change the figure caption “Figure 1. SEM images of starting powders. (a) Raw Al powder, (b) Al powder after 3h ball milled, (c) Al powder after 8h ball milled, (d) Al powder after 20h ball milled.

(ii) Please change Figure 2 and provide clear image”.

(iii) Please change the figure caption of Figure 4.

(iv) Please provide the Vickers hardness graph instead of table.

(v)  Please provide clear graph of Figure 5.

RECOMMENDATION

After reviewing the enclosed manuscript for “Materials”, the present manuscript contains some kinds of scientific analysis but it is mandatory required to modify according to the preceding remarks. So, the manuscript can be accepted for publication after major mandatory revisions have been made.

Author Response

Thank you very much for giving us an opportunity to revise our manuscript. We appreciate the worthy and constructive comments and suggestions on our manuscript originally entitled “Study on in-situ synthesis process of Ti-Al intermetallic compounds reinforced Al matrix composites” (materials-517469). The comments are precious for the authors to revise and improve the manuscript. We have studied the comments carefully and revised the manuscript in detail. All the changes in the revised manuscript were highlighted.

Yours sincerely,

Wan Qiong

------------------------------------------------------------------------------------

Reviewer1
1Title: Study on in-situ synthesis process of Ti-Al intermetallic compounds reinforced Al matrix composites with core-shell-like structure

Replaced by

Study on in-situ synthesis process of Ti-Al intermetallic compounds reinforced Al matrix composites

 Response: Thank you for your worthy comments. This problem has been corrected and marked as red.

2Abstract:

-----Ti-Al intermetallic compounds reinforced aluminum matrix composites

Replaced by

---Ti-Al intermetallic compounds reinforced aluminum (Al) matrix composites---"

After that Please replaced throughout the manuscript aluminum to Al

 Response: Thank you for your worthy comments. This problem has been corrected and marked as red.

----“ The specimens were observed to study the microstructure of reinforcement by OM, EDS and EBSD. -”

Replaced by

---- A detail microstructural characterization of Ti-Al intermetallic compounds reinforced Al composite was carried out by OM, EDS and EBSD

 Response: Thank you for your worthy comments. This problem has been corrected and marked as red.

3Introduction part:

“In recent years, many researchers developed ceramic particle reinforced aluminum matrix composites, which were used in the automobile industry and manufacture of other structural parts. The widely used ceramic particles are SiC, Al2O3, TiB2, B4C and TiC [4-8]”

Replaced by

“In recent years, many researchers developed ceramic particle reinforced metal matrix composite, which were widely used in the automobile industry and structural applications. The widely used ceramic particles are SiC, ZrO2, Al2O3, TiB2, B4C and TiC -----[add ref.]”

1. A.K. Gain, L Zhang, Microstructure, mechanical and electrical performances of zirconia nanoparticles-doped tin-silver-copper solder alloys, J. Mater. Sci.: Mater. Electron. 27(7), (2016) 7524-7533.

2.  A.K. Gain, L Zhang, M.Z. Quadir, “Composites matching the properties of human cortical bones: the design of porous titanium-zirconia (Ti-ZrO2) nanocomposites using polymethyl methacrylate powders” Mater. Sci. Eng. A 662 (2016) 258-267.

 Response: Thank you for your worthy comments. This problem has been corrected and marked as red.

Please add references “The intermetallic compounds have low density, high melting point, high strength and good resistance to corrosion and creep.”

Response: Thank you for your worthy comments. This problem has been corrected and marked as red.

Please add the novelty and aim of this work in page 2 line 91-93

Response: Thank you for your worthy comments. This problem has been corrected and marked as red.

4Experimental procedure part:

Please include all equipment name and model number.

Response: Thank you for your worthy comments. This problem has been corrected and marked as red.

(5) Results and discussion:

(i)                 Figure 1: Please change the figure caption “Figure 1. SEM images of starting powders. (a) Raw Al powder, (b) Al powder after 3h ball milled, (c) Al powder after 8h ball milled, (d) Al powder after 20h ball milled.

Response: Thank you for your worthy comments. The figure caption has changed to be “SEM images starting powders. (a) Raw Al powder, (b) blended powder after 3h ball milled, (c) blended powder after 8h ball milled, (d) blended powder after 20h ball milled.” Because figure (b), (c) and (d) are blended powders of Ti and Al.

(ii)               Please change Figure 2 and provide clear image”.

Response: Thank you for your worthy comments. The resolution of images has been improved.

(iii)             Please change the figure caption of Figure 4.

Response: Thank you for your worthy comments. The figure caption of Figure 4 has been changed as “Figure 4. (a) SEM image and (b) EBSD analysis of specimen C.” and marked as red in the paper.

(iv)             Please provide the Vickers hardness graph instead of table.

Response: Thank you for your worthy comments. The Vickers hardness graph has been instead of table.

(v)       Please provide clear graph of Figure 5.

Response: Thank you for your worthy comments. Figure 6 in the paper now has been provided with high revolution.

RECOMMENDATION

After reviewing the enclosed manuscript for “Materials”, the present manuscript contains some kinds of scientific analysis but it is mandatory required to modify according to the preceding remarks. So, the manuscript can be accepted for publication after major mandatory revisions have been made.

Thank you for your worthy comments.

Reviewer 2 Report

Dear Authors,

The reviewed paper "Study on in-situ synthesis process of Ti-Al 2 intermetallic compounds reinforced Al matrix 3 composites with core-shell-like structure" describes a processing of in-situ formation of Al/TiAl MMC. Materials and experimental details are described properly. The applied technique is described and discussed properly.

On the other hand, no overall change of material's density resulted by Ti-Al interface reaction has not been mentioned and discussed. That leaded to unsupported concluding regarding Kirkendall effect, which actually cannot be accepted or rejected. No any other microstructure characterization technique (for example, XRD) has  been applied/presented. 

On my opinion, the good work done by yourself should be a part of significantly deeper and comprehensive future study which will be much more scientifically interesting for the reader, than the present paper.   

Author Response

Thank you very much for giving us an opportunity to revise our manuscript. We appreciate the worthy and constructive comments and suggestions on our manuscript originally entitled “Study on in-situ synthesis process of Ti-Al intermetallic compounds reinforced Al matrix composites” (materials-517469). The comments are precious for the authors to revise and improve the manuscript. We have studied the comments carefully and revised the manuscript in detail. All the changes in the revised manuscript were highlighted.

Yours sincerely,

Wan Qiong

------------------------------------------------------------------------------------

Reviewer2

The reviewed paper "Study on in-situ synthesis process of Ti-Al 2 intermetallic compounds reinforced Al matrix 3 composites with core-shell-like structure" describes a processing of in-situ formation of Al/TiAl MMC. Materials and experimental details are described properly. The applied technique is described and discussed properly.

(1)On the other hand, no overall change of material's density resulted by Ti-Al interface reaction has not been mentioned and discussed. That leaded to unsupported concluding regarding Kirkendall effect, which actually cannot be accepted or rejected.

Response: Thank you for your worthy comments. Kirkendall effect is caused by the difference of diffusion coefficient between Al and Ti. In cold pressing sintering process, the cylindrical billet with a diameter of 16 mm was consolidated under the load of 20 tons. The high pressure applied on the billet made the powder particles compact closely. When the billet was sintered, the diffusion distance required by in-situ reaction is short. Once the intermetallic shell structure formed, it impeded the diffusion and reduced the Kirkendall effect. While in hot pressing sintering process, the diffusion became more and more active with the temperature rising, the place initially occupied by Al was empty because the diffusion of Al to Ti is predominant. Although the voids could be diminished under the pressure, the pressure is not high enough to coalesce the voids around reinforcements and the interspace between cores and cells. In addition, the radiate cracks in the shell structure promote the diffusion during sintering process. Therefore, many voids remained in Al matrix composites. Based on the above analysis and your suggestion, the conclusion “The Kirkendall effect caused by the difference in diffusion coefficient results in the formation of cavities and the decrease of density degree” was made for hot pressing sintering process in this paper.

(2)No any other microstructure characterization technique (for example, XRD) has been applied/presented.

Response: Thank you for your worthy comments. XRD is widely used in the phase analysis and structure determination. In this paper, the shell-core structure in reinforcement is the study emphasis. So EBSD and linear scan were selected to determine the constituent and distribution of intermetallic compounds formed in composites.

(3)On my opinion, the good work done by yourself should be a part of significantly deeper and comprehensive future study which will be much more scientifically interesting for the reader, than the present paper. 

Response: Thanks for your approval on my work. The significantly deeper and comprehensive study will be carried out. Based on the research work of this paper, more high-quality research results will be made in the future.

Reviewer 3 Report

This paper is well-organized and written with proper English. Results are well-discussed and useful. Some specific comments are given below.

What is the content (the molar ratio or weight ratio) of Ti-Al intermetallics formed in the Al matrix?

Is it possible to identify the Ti-Al compounds by XRD analysis? It will also be useful to provide the EDS spectrum and atomic ratio of the Ti-Al phase.

It is useful to provide the fracture toughness of the Ti-Al intermetallics reinforced Al samples.

The holding time was shown to be the more important factor. Is this conclusion true for the hot pressing sintering case?

Author Response

Thank you very much for giving us an opportunity to revise our manuscript. We appreciate the worthy and constructive comments and suggestions on our manuscript originally entitled “Study on in-situ synthesis process of Ti-Al intermetallic compounds reinforced Al matrix composites” (materials-517469). The comments are precious for the authors to revise and improve the manuscript. We have studied the comments carefully and revised the manuscript in detail. All the changes in the revised manuscript were highlighted.

Yours sincerely,

Wan Qiong

------------------------------------------------------------------------------------

Reviewer3

Comments and Suggestions for Authors

This paper is well-organized and written with proper English. Results are well-discussed and useful. Some specific comments are given below.(1) What is the content (the molar ratio or weight ratio) of Ti-Al intermetallics formed in the Al matrix?

Response: Thank you for your worthy comments. According to the Ti-Al phase diagram, there are a series of Ti-Al intermetallics can be formed in the Al matrix, including Ti3Al, TiAl, TiAl2 and TiAl3. Among these intermetallics, the density of TiAl3 is the smallest and its resistance to oxidation is excellent, so TiAl3 was taken as the main reinforcement in this study. The main intermetallic compounds TiAl3 were in-situ formed in the Al matrix composites by adjusting the process parameters of ball milling and sintering in this paper.

(2) Is it possible to identify the Ti-Al compounds by XRD analysis? It will also be useful to provide the EDS spectrum and atomic ratio of the Ti-Al phase.

Response: Thank you for your worthy comments. XRD analysis can identify the Ti-Al compounds and provide atomic ratio of the Ti-Al phase. The results can indicate the intermetallics formed in Al matrix. The structure of the reinforcement and the distribution of these intermetallics are the focus of this study, so EBSD was selected to identify the Ti-Al compounds and determine the core-cell structure of reinforcement.

 (3) It is useful to provide the fracture toughness of the Ti-Al intermetallics reinforced Al samples.

Response: Thank you for your worthy comments. Just as the reviewer pointed out, characterization on the fracture toughness of the Ti-Al intermetallics reinforced Al samples is a significant thing. Because the samples fabricated in the present study are small, the specific analysis on fracture toughness will be made in the subsequent research.

 (4)The holding time was shown to be the more important factor. Is this conclusion true for the hot pressing sintering case?

Response: Thank you for your worthy comments. In our present study, the rotation speed in ball milling is below 300rmp (the rotation speed is 150 rmp in this paper), which belongs to the range of low energy ball milling. The energy provided by ball milling can only promote the mechanic bonding between the particles. The chemical bonding in intermetallic compounds was formed in the holding process or hot pressing sintering process. Therefore, the milling time was the less important factor, and the holding time was shown to be the more important factor. This conclusion is true for the hot pressing sintering case.

Round  2

Reviewer 2 Report

Dear Authors,

In order to bring you paper into the publishing-ready condition, I kindly ask you to provide the following information:

To add X-rays diffractometry for additional confirmation of TiAl formation (SEM/EDS is insufficient).

Vickers hardness measurement: imprint size (in order to compare with TiAl particles size). 

Quantitative estimation of the overall density reduction due to formation of intermetallic compound - in order to compare the volume of the formed porosity and then to be able to conclude regarding Kirkendall effect (exists or not).

On my opinion, throughout fulfillment of the mentioned requirements, the paper may be reviewed and then - depending on the review's results - published. 

Author Response

Thank you very much for giving us an opportunity to revise our manuscript. We appreciate the worthy and constructive comments and suggestions on our manuscript originally entitled “Study on in-situ synthesis process of Ti-Al intermetallic compounds reinforced Al matrix composites” (materials-517469). The comments are precious for the authors to revise and improve the manuscript. We have studied the comments carefully and revised the manuscript in detail. All the changes in the revised manuscript were highlighted in blue.

Yours sincerely,

Wan Qiong

------------------------------------------------------------------------------------

Reviewer 2

In order to bring you paper into the publishing-ready condition, I kindly ask you to provide the following information:

(1) To add X-rays diffractometry for additional confirmation of TiAl formation (SEM/EDS is insufficient).

Response: Thank you for your worthy comments. The XRD spectra have been provided to confirm the TiAl formation in Fig. 2.

 (2) Vickers hardness measurement: imprint size (in order to compare with TiAl particles size). 

Response: Thank you for your worthy comments. The mean values of two diagonals in Vickers hardness measurement are between 0.28-0.51mm approximately. While most TiAl particles size is about 50μm (0.05mm) and few particle size reaches 100μm (0.1mm). It indicates both TiAl particles and Al matrix are under the load of indenter in Vickers hardness measurement.

 (3) Quantitative estimation of the overall density reduction due to formation of intermetallic compound - in order to compare the volume of the formed porosity and then to be able to conclude regarding Kirkendall effect (exists or not).

Response: Thank you for your worthy comments. The Kirkendall effect is a classical phenomenon in metallurgy [1-3]. It basically refers to a nonreciprocal mutual diffusion process through an interface of two metals so that vacancy diffusion occurs to compensate for the unequality of the material flow and that the initial interface moves [4]. Many researchers have verified the existence of Kirkendall effect in the diffusion reaction of solid Ti and Al [5-8] and used this effect to fabricate hollow or porous materials [9-11]. The overall density reduction is affected by pressure, sintering temperature, the particle size of as-milled powders and other factors in the fabrication [12]. Whether in cold pressing sintering or hot pressing sintering, the gases were inevitably included in the cylinder billet and hot pressing mould. If these gases can’t be released completely, the pore will retain in the composites, which can also reduce the overall density of the material. Therefore, the qualitative analysis on density reduction was made in this paper. It is very regretful that the overall density reduction due to formation of intermetallic compound, process parameters or Kirkendall effect could not the quantitative estimated at our present study. But we will make this problem as one of our research directions in the future.

References

[1] a) E. O. Kirkendall, Trans. AIME 1942, 147, 104; b) Smigelskas A D, E. O. Kirkendall, Trans. AIME 1947, 171, 130.

[2] Nakajima H. The discovery and acceptance of the Kirkendall Effect: The result of a short research career[J]. J. Miner. Met. Mater. Soc. 1997, 49(6), 15-19.

[3]Paul A. The Kirkendall effect in solid state diffusion. PhD Thesis, Technische Universiteit Eindhoven, The Netherlands, 2004

[4] Fan H J, Gösele U, Zacharias M. Formation of Nanotubes and Hollow Nanoparticles Based on Kirkendall and Diffusion Processes: A Review[J]. Small, 2007, 3(10):1660-1671.

[5] Kulkarni K N, Sun Y, Sachdev A K, et al. Field-activated sintering of blended elemental γ-TiAl powder compacts: Porosity analysis and growth kinetics of Al3Ti[J]. Scripta Materialia, 2013, 68(11):841-844.

[5] He Y, Jiang Y, Xu N, et al. Fabrication of Ti-Al Micro/ Nanometer-sized Porous Alloys through the Kirkendall Effect[J]. Advanced Materials, 2007, 19(16):2102-2106.

[6] Novoselova T, Celotto S, Morgan R, et al. Formation of TiAl intermetallics by heat treatment of cold-sprayed precursor deposits[J]. Journal of Alloys and Compounds, 2007, 436(1-2):69-77.

[7] Sienkiewicz J, Kuroda S, Minagawa K, et al. Effects of Al Content and Addition of Third Element on Fabrication of Ti-Al Intermetallic Coatings by Heat Treatment of Warm-Sprayed Precursors[J]. Journal of Thermal Spray Technology, 2015, 24(5):749-757.

[8] Rezaei A, Madaah Hosseini H R. Evolution of microstructure and mechanical properties of Al-5wt% Ti composite fabricated by P/M and hot extrusion: Effect of heat treatment[J]. Materials Science and Engineering: A, 2017, 689:166-175.

[9] Yin Y, Rioux R M, Erdonmez C K, et al. Formation of Hollow Nanocrystals Through the Nanoscale Kirkendall Effect[J]. Science, 2004, 304(5671):711-714.

[10] He Y, Jiang Y, Xu N, et al. Fabrication of Ti-Al Micro/ Nanometer-sized Porous Alloys through the Kirkendall Effect[J]. Advanced Materials, 2007, 19(16):2102-2106.

[11] Nakajima H, Nakamura R. Diffusion in Intermetallic Compounds and Fabrication of Hollow Nanoparticles through Kirkendall Effect[J]. Journal of Nano Research, 2009, 7(1):1-10.